# Plasmonic Metal Nanoparticles Hybridized with 2D Nanomaterials for SERS Detection: A Review

**DOI:** 10.3390/bios12040225

**Published:** 2022-04-09

**Authors:** Caterina Serafinelli, Alessandro Fantoni, Elisabete C. B. A. Alegria, Manuela Vieira

**Affiliations:** 1Instituto Superior de Engenharia de Lisboa—Instituto Politécnico de Lisboa, 1949-014 Lisboa, Portugal; afantoni@deetc.isel.ipl.pt (A.F.); ebastos@deq.isel.ipl.pt (E.C.B.A.A.); mv@isel.ipl.pt (M.V.); 2Centro de Química Estrutural, Institute of Molecular Sciences, Instituto Superior Técnico, Universidade de Lisboa, 1049-001 Lisboa, Portugal; 3CTS—Centre of Technology and Systems, Caparica, 2829-516 Almada, Portugal; 4Department of Electrotechnical and Computer Engineering, Faculty of Science and Technology, Universidade NOVA de Lisboa, DEE-FCT-UNL, Caparica, 2829-516 Almada, Portugal

**Keywords:** SERS analysis, plasmonic metal nanoparticles, hotspots, hybrid materials

## Abstract

In SERS analysis, the specificity of molecular fingerprints is combined with potential single-molecule sensitivity so that is an attractive tool to detect molecules in trace amounts. Although several substrates have been widely used from early on, there are still some problems such as the difficulties to bind some molecules to the substrate. With the development of nanotechnology, an increasing interest has been focused on plasmonic metal nanoparticles hybridized with (2D) nanomaterials due to their unique properties. More frequently, the excellent properties of the hybrids compounds have been used to improve the drawbacks of the SERS platforms in order to create a system with outstanding properties. In this review, the physics and working principles of SERS will be provided along with the properties of differently shaped metal nanoparticles. After that, an overview on how the hybrid compounds can be engineered to obtain the SERS platform with unique properties will be given.

## 1. Introduction

The SERS technology fits very well in the scenario of the rapidly emerging new technologies, realizing the ultimate goal of analytical chemistry: the detection, analysis, and manipulation of single molecules, namely the single-molecule detection. For example, a growing interest is focused on groundbreaking “single-molecule electrical approaches” methods that translate chemical or physical processes into detectable electrical signals at the single-event level on the platform of single-molecule electronic devices [1]. Among the different strategies developed in recent years, graphene−molecule−graphene single-molecule junctions (GMG-SMJs) are particularly attractive, showing great potential for the routine applications [2]. Other approaches to reach single-molecule sensitivity are focused on mechanical strategies such as Atomic Force Microscopy (AFM). Owing to the atomically well-defined tip apex and its mechanical flexibility, AFM has revealed an intriguing potential to directly characterize the molecular structure in real space with an outstanding single-molecule sensitivity, and on these bases, a new generation of AFM, the single-molecule AFM (sm-AFM), has been developed for the characterization and manipulation of single molecules [3]. Other approaches are based on optical strategies: for example, new fluorescence microscopy techniques surpassing the diffraction limit of the traditional optical microscopes are receiving a great interest. The development of super-resolved fluorescence microscopy to achieve “super-resolution” led to wide applications in many scientific fields and was recognized by the Nobel Prize in Chemistry in 2014. Other approaches are based on LSPR of plasmonic nanoparticles [4]. A small overview has been depicted on the techniques aiming to reach the single-molecule sensitivity, but there is still a lot of work to do for their massive use as commercial biosensing platforms. Due to its great potential to be implemented in commercial use, in recent years a growing interest has been concentrated in SERS analysis, so that on the basis of extensive research being produced to go deep inside the mechanisms and in its applications, numerous reviews have been published with the aim to help the researchers in the next steps of their studies [5,6]. However, despite the large number of works published covering a wide range of topics, only a few works have been focused on the use of hybrid structures composed of plasmonic metal nanoparticles and 2D nanomaterials in SERS analysis [7]. Stimulated by the enormous progress made on the knowledge and applications of the hybrid compounds, in this review we highlight their use in SERS analysis in order to fill the gap in this direction. The following work is organized as follows: in Section 1, an introduction illustrating the scenario of SERS analysis has been presented, whereas in Section 2, physics and working principles of the SERS technique are provided. In Section 3, the work on the early use of hybrid structures is addressed, focusing attention on spherical plasmonic metal noble nanoparticles and graphene or its derivates in SERS analysis. In Section 4, anisotropy in plasmonic metal nanoparticles is introduced and it is explained how the performances in SERS analysis are enhanced. The use of anisotropic metal nanoparticles in hybrid compounds to improve the performance of SERS analysis is depicted in Section 5. After discussing the effect of particle anisotropy in hybrid compounds, in the following sections, the different approaches to improve the performance of the SERS technique will be highlighted. In Section 6, the engineering of the bidimensional material is discussed, while considering how the hybrid compounds are evolving towards three-dimensional structures with the aim of improving the SERS properties even more in Section 7. The use of bidimensional material as a nano spacer is presented in Section 8, taking into account the use of plasmonic metal nanoparticles veiled with bidimensional material in Section 9. Considering different directions in which the research of SERS is focused, a general overview on the future of SERS will be provided in Section 10, along with the conclusion in Section 11.

## 2. Physics and Working Principles

Raman spectroscopy is a valuable technique to study chemical and intramolecular bonds by producing and examining inelastic scattering generated from molecules, thus providing a vibrational fingerprint, unique for each molecule, becoming a powerful tool to identify chemical species, supplying both qualitative and quantitative molecular information from any sample. Unfortunately, this is a very weak process, but it has been found that when the molecules are located near a rough metal surface or metal nanoparticles (NP), a Raman scattering boost occurs, greatly enhancing the signal intensity [8], thus opening the way to a new emerging research field: the surface-enhanced Raman scattering (SERS) spectroscopy, that has proved to be a powerful technique for non-invasive, rapid and reliable sensing of chemicals and biomolecules [9,10,11]. The Raman enhancement in SERS spectroscopy occurs when an incident electromagnetic field interacts with surface plasmon resonance at a metal surface (SERS effect) [12]. The unique optical properties of metal nanostructures are related to the presence of strong localized plasmon resonances (LSPR), excitation of coherent, collective oscillations of delocalized electrons in the conduction band by an external electromagnetic (EM) field as the driving force [13]. When metal nanostructures interact with a light beam, part of the incident photons are absorbed, and part are scattered in different directions: both absorption and scattering are greatly enhanced when the LSPR is excited [14]. The LSPR generated by metal nanostructures produces extremely intense and highly confined electromagnetic fields within the gaps between metallic nanostructures termed “hotspots” [15] that have been claimed to provide extraordinary enhancements of up to 10^15^ orders of magnitude to the surface-enhanced Raman scattering (SERS) signal [16]. Generally accepted mechanisms of SERS enhancement are attributed to two effects: the first one is an electromagnetic mechanism (EM) related to the striking increase in the local electromagnetic field near the metal nanoparticle surface, whereas the second one is a chemical mechanism (CM) involving a charge transfer [17]. The EM is a process originated from the electromagnetic interaction between the metal nanoparticles and the molecules, implicating two mechanisms: the first is the result of the interaction of the metal nanoparticles with the incident beam, whereas the second one is a re-radiation phenomenon [18]. When a metal nanoparticle interacts with an incident field *E*_0_ at a wavelength *λ*_0_ surface plasmon resonance (SPR), oscillating dipoles are induced and the resulting polarization generates large fields around the particle. The enhanced local field *E_loc_* around the metal nanoparticles generated by the interaction with incident light is proportional to the incident field *E*_0_ and to a factor called local enhancement factor *M_loc_* (*λ*_0_) and can be quantified as:*E_loc_* = *E*_0_
*M_loc_* (*λ*_0_)(1)

The enhanced local field *E_loc_* excites the molecule inducing a dipole so that the molecule scatters the Raman signal in all directions radiating an enhanced scattered field *E_scat_* at a Raman wavelength *λ_r_* shifted from *λ*_0_ with an intensity proportional to the molecule polarizability *α* and the enhanced local field *E_loc_*.
*E_scat_* = *α*
*E_loc_* = *α*
*E*_0_
*M_loc_* (*λ*_0_)(2)

In these circumstances, the Raman signal is already enhanced with respect to the case in which molecules do not undergo the presence of the nanoparticle at its vicinity. On the second step, the scattered field *E_scat_* interacts with the metal nanoparticle and as a result, it is enhanced by a re-radiation process that is expressed by:*E_rad_* = *E_scat_*
*M_rad_* (*λ_r_*) = *α*
*M_rad_* (*λ_r_*) *M_loc_* (*λ*_0_) *E*_0_(3)

The chemical enhancement is a process originated from a change in the molecule polarizability (along with the Raman cross-section of its vibrational modes), resulting from the physicochemical interaction between the substrate and the molecule. When the molecule interacts with the substrate by means of physisorption or chemisorption, its geometrical and electronic structures are modified, so that the Raman cross-section of its vibrational modes is different compared to that of the free molecule. The chemical enhancement arises from two different mechanisms. The first one is the *non-resonant chemical effect*: when the molecular orbitals have energies far from the Fermi level of the metal, new electronic states are not formed but the electronic and geometrical structures of the molecule are transformed, resulting in the modification of the Raman shifts and the intensity of the vibrational modes. The second mechanism is a *resonant charge transfer chemical effect*: the interaction between the metal and the molecule leads to a metal–molecule charge transfer state (CT) and in the case in which the laser source is in resonance with the CT state, the Raman modes are strongly enhanced. In addition, the chemical effect can also originate from a temporary electron transfer between the molecule and the metal, a “transient” charge transfer [19]. Although amplified by both EM and CM, SERS efficacy greatly benefits from EM enhancements, with a contribution of 10^8^ or more, whereas the contribution of the chemical effect would not exceed a factor 100, thus suggesting that the EM enhancement is the dominant contribution to the SERS sensitivity [20]. Figure 1 presents a schematic showing the SERS signal arising from a molecule localized in the hotspot formed in the junction between two gold nanoparticles.

A key parameter in quantifying the overall signal increasing is the SERS enhancement factor (*EF*) that is experimentally evaluated by means of SERS intensity measurements for the adsorbed molecule on the metal surface, relative to the normal Raman intensity of the same, “free” molecule in the solution. One of the most used equations to calculate *EF* is expressed by:(4)$$EF=\frac{I_{SERS}/N_{Surf}}{I_{RS}/N_{Vol}}$$
where *N_Vol_ = c_RS_V* is the average number of molecules in the scattering volume (V) for the Raman (non-SERS) measurement, and *N_Surf_* is the average number of adsorbed molecules in the scattering volume for the SERS experiments while *I_SERS_* and *I_RS_* are the intensities of the same band for the SERS and bulk spectra [21].

## 3. Hybrid Nanocomposites

Plasmonic nanoparticles, in condition of resonance, are able to generate a strong electromagnetic field [22] and in the past they have been exploited as excellent SERS substrates, mainly due to their huge enhancement induced by the EM effect. Despite the huge potential of SERS analysis, the low affinity of non-thiolated molecules to metals, such as gold or silver, is still a challenge because the affinity determines the retention of analytes. To bypass this complication, several approaches have been developed. For example, core-shell colloidal material comprising gold nanoparticles coated with a thermally responsive poly-(N-isopropylacrylamide) (pNIPAM) (Au@pNIPAM) has been developed as a SERS substrate [23]: the plasmonic metal core provides the enhancing properties, whereas the pNIPAM shell is exploited to trap and get the molecules close to the metal core to obtain the SERS signal. Other applications include other coatings such as calixarenes [24]. In this context, bidimensional (2D) nanomaterials such as graphene and its derivatives, 2D metallic oxides, hexagonal boron nitride (h-BN)) etc., have attracted great interest and have been investigated as SERS substrates for their intriguing properties such as an enhanced photogeneration rate, the plasmon-induced ‘‘hot electrons’’ and improved conductivity [25]. Between 2D nanomaterials, transition metal dichalcogenides (TMDCs) is a large family of materials of MX_2_ type, where M is a transition metal element from group IV, V or VI, for example, Mo or W and X is a chalcogen (S, Se or Te). They possess a layered structure, and each layer includes three atomic planes with an arrangement of X-M-X type, where a hexagonally packed plane of transition metal atoms M is enclosed within two atomic planes of chalcogen X resulting in a thickness of 6–7 Å for each single layer [26]. Inside the TMDCs family, MoS_2_ and Tungsten disulfide (WS_2_) received great interest because of their properties and have been exploited to produce advanced SERS substrates. Graphene and its derivative such as graphene oxide (GO) or reduced graphene oxide (r-GO) have been the most extensively investigated and have emerged as a material for SERS substrates, due to their intriguing properties. Firstly, its unique structure is favorable for interactions with analytes via π-π stacking and hydrophobic interactions thus facilitating the adsorption of non-thiolated molecules, resulting in an enhancement of the SERS signal. Secondly, it has been shown that it contributes to the SERS signal enhancement with a magnitude depending on the degree of GO chemical reduction [27]. The signal SERS enhancement is dominated by chemical mechanism [28] rather than the electromagnetic mechanism, although the EF is not dramatic as in the case of metal substrates. Moreover, graphene offers the additional advantage of fluorescence quenching, improving the SERS efficacy. Larger cross-sections for the fluorescent signal, compared to Raman signal, are observed, so that Raman characteristics are often interfered with, or even submerged, by the intense fluorescence background, thus lowering the quality of SERS analysis. A fluorescence quenching effect of fluorescent dyes rhodamine 6G (R6G) and protoporphyrin IX (PPP) adsorbed on graphene was first reported by Zhang [29]. The approximate evaluated quenching factor has been found on the order of 10^3^ and the quenching effect has been assigned to a resonance energy transfer process according to the results obtained by Swathi [30]. The chemical enhancement on three types of different 2D nanomaterials, graphene, hexagonal boron nitride (h-BN) and molybdenum disulfide (MoS_2_) (each having different electronic properties) has been investigated by Ling et al. [31] by means of the copper phthalocyanine (CuPc) molecule as a probe. It has been observed that different vibrational modes showed different enhancement factors depending strongly on the substrates. These inconsistencies have been related to three different enhancement mechanisms determined by the distinct electronic properties and chemical bonds exhibited by the three substrates. Graphene is zero-gap semiconductor and has a nonpolar C-C bond [32], so that the Raman enhancement is assigned to the strong charge transfer with the CuPc molecule and not to the weak dipole-dipole interactions due to the nonpolar nature of graphene. Differently, h-BN is highly polar (due to the strong B-N bond) and insulating with a large band gap (5.9 eV) [33] so that the dipole interactions are dominant while the charge transfer interactions are negligible, and the signal amplification results from dipole-dipole interactions between h-BN and CuPc molecules. In addition, MoS_2_ is semiconducting [34] and less polar compared to h-BN so that, although both dipole-dipole interactions and charge transfer occur, they are weaker thus resulting in a weaker signal enhancement. In view of the huge progress in designing and developing plasmonic metal nanoparticles as SERS substrates originating from their EM enhancement, and in the use of graphene and bidimensional nanomaterials as effective Raman enhancement substrates due to their CM, further advances in reaching greater performances in SERS analysis have been obtained by the combination of plasmonic metal nanoparticles with 2D nanomaterials. The obtained hybrid composites displayed advanced enhancement properties, arising from the synergistic effect of the EM (originating from the high local electric field at hotspots formed in metallic NPs) and CM (deriving from the charge transfer between 2D nanomaterials and probe molecule). In the early works, hybrid composites have been produced by growing spherical metal nanoparticles on graphene nanosheets. A hybrid system (GO/PDDA/AgNPs) has been developed, self-assembling Ag spherical nanoparticles on the surface of graphene oxide by means of poly (diallyldimethyl ammonium chloride) (PDDA) and successively tested as SERS platform sensing for folic acid detection, showing strong SERS activity. The SERS spectra of p-aminothiophenol (p-ATP) collected using Ag colloid and GO/PDDA/AgNPs showed a greater enhancement signal for spectrum obtained with Ag colloid due to the fact that GO did not assist the adsorption of p-ATP on AgNPs of GO/PDDA/AgNPs. In the case of folic acid detection, the SERS signals on GO/PDDA/AgNPs were much stronger than that on Ag nanoparticles [35]. Stimulated by the striking properties arising from the synergistic effect of EM and CM in the hybrid nanocomposites, Chen [36] developed a SERS platform based on p-aminothiophenol (PATP)-functionalized silver nanoparticles supported on graphene nanosheets (Ag/GNs) for the sensitive and selective detection of 2,4,6-trinitrotoluene (TNT). As a first step, graphene nanosheets were decorated with silver nanoparticles by reducing silver nitrate with sodium citrate and then the Ag/GNs composites have been functionalized with p-aminothiophenol (PATP) obtaining the PATP-Ag/GNs composites. In presence of TNT, π-π conjugated structures between TNT and PATP are created, promoting the effective charge transfer from the electron-rich PATP to the electron-poor TNT that leads to the enhanced Raman signals. With a LOD of 5.0 × 10^−16^ M, the PATP-Ag/GNs hybrid displayed a great sensitivity towards TNT detection. A SERS substrate has been prepared according to the one-step strategy in the work of Wei [37]. Hybrid structures have been produced by simultaneous reduction of GO and HAuCl_4_ with sodium citrate and ammonia, and then an RGO/AuNP film has been deposited on a silicon wafer and on a poly (ethylene terephthalate) (PET) substrate to create the SERS platform. To explore the potential of the created RGO/AuNP composite, 4-aminothiophenol (4-ATP) has been exploited as probe molecule in SERS analysis, finding a value of 5.6 × 10^5^ for the enhancement factor (EF), such that the RGO/AuNP hybrid displayed remarkable performances in 4-ATP detection. On the basis of the results obtained, the composite has been exploited as a SERS substrate to detect 2-thiouracil (2-TU) with a low concentration to 1 μM, thus confirming the great ability as a biodetection platform. However, due to their attractive properties, not only graphene has been exploited to create hybrid structures with plasmonic metal nanoparticles, but several types of 2D nanomaterials have also been used to produce composites to be utilized as SERS substrates. A SERS active substrate has been constructed by Chao [38] by growing Au nanoparticles on Molybdenum disulfide (MoS_2_) nanosheets. The gold precursor has been directly reduced on the MoS_2_ surface in the presence of carboxymethyl cellulose (CMC) as a stabilizer in an aqueous solution to create the AuNPs@MoS_2_ nanocomposite. Several AuNPs@MoS_2_ nanocomposites have been prepared with different amounts of Au nanoparticles and the one with the best performance has been selected by means of the standard probe rhodamine 6G (R6G), finding a value of 8.2 × 10^−5^ for the enhancement factor (EF). The amplification of the SERS signal has been assigned to the hot spots generated by the little aggregation and closeness of Au nanoparticles. An approach to exploit the unique properties of a nanohybrid formed by gold nanoparticles (AuNPs) deposited onto exfoliated nanosheets of tungsten disulfide (WS_2_) was presented in the work of Sabherwal [39]: an active SERS platform based on an Au NPs/WS_2_ nanohybrid has been developed for the label-free detection of Myoglobin, a cardiac biomarker. The AuNPs/WS_2_ nanohybrid has been prepared by the in situ reduction of gold salt precursor, and then the surface was functionalized with specific aptamers to impart high selectivity towards Myoglobin. The prepared nanohybrid has been tested for SERS detection. The obtained results showed the synergistic use of the unique properties of chemical and electromagnetic enhancement of both WS_2_ and AuNPs for a many fold increase in the SERS signal intensity. The AuNPs/WS_2_ nanohybrid system allowed the Myoglobin detection with a LOD of 10^−2^ pg mL^−1^, considerably lower with respect to that measured in other works. In Table 1, the results obtained for SERS analysis using hybrids of plasmonic metal nanoparticles and graphene or other 2D nanomaterials are listed.

## 4. Effect of Nanoparticle Shape

Plasmon resonances in spherical nanoparticles can be tuned in a limited range of wavelengths by changing the particle diameter, while introducing anisotropy in the particle shape provides a fine control of plasmon resonance thus broadening the range of wavelengths from the visible through the mid IR, by varying the aspect ratio (AR) of the NPs [40]. On these presuppositions, the early attempts to further improve the SERS efficacy of the hybrid composites exploited the engineering of particle morphology tailoring the particle anisotropy along with the plasmonic properties and intrinsic electromagnetic ‘‘hotspots’’.

### 4.1. Au Nanorods

Rod-shaped nanoparticles are nanostructures where one dimension is longer than the other two, so that the term nanorods indicates elongated nanoparticles. They exhibit two plasmon resonances bands: the first one is longitudinal mode parallel to the long rod axis, whereas the second one is a transverse mode perpendicular to the long axis of the rod [41]. Different from the transverse band and characterized by a low intensity and independence from the aspect ratio (AR), the longitudinal mode is much more intense and strongly independent from AR, so that when controlling the AR of AuNRs, it is possible to tune the plasmon resonance across the UV-visible region of the spectrum [42]. From theoretical discrete dipole approximation (DDA) and experimental electron energy loss spectroscopy (EELS), near-field maps illustrated an intense EM field in the proximity of the AuNRs originating from plasmon modes [43,44]. From the calculations, results show a high electromagnetic (EM) field enhancement at the Au nanorods’ tips (hotspots) under longitudinal excitation, whereas the enhancement is moderate on the NR lateral sides for transverse excitation. In addition, the AR also controls the SERS efficacy of the AuNRs: exploiting crystal violet (CV) as a molecule probe, it has been shown that the EF increases when the AR values become greater, so that controlling the AuNRs’ aspect ratio enables a fine tuning of plasmon resonance and the EF for an optimized SERS analysis [45]. For the sake of having a more complete view on the properties of elongated particles, it is worth mentioning that, with a tight and precise control over the synthesis conditions, other elongated structures have been obtained, such as gold nano bipyramids. Depending on their sharp nanotips, gold nano bipyramids (AuNBPs) revealed a stronger local field enhancement compared to AuNRs [46]. Due to the sharp tips, in AuNBs the radiation can be centralized into a strong local electric field and enhancing at the same time the local density of photonic states [47], resulting in the beneficial application for spectroscopy, photocatalysis, detection and biomedicine [48].

### 4.2. Au Nanotriangles

The term nanoplates indicates nanoparticles in which one dimension is much smaller than the other two, and the specific case in which the base is triangular, the nanoparticles are referred as nanotriangles (NTs). Nanotriangles are characterized by intriguing properties originating from the morphology: the EM fields can be confined by their intrinsic sharp corners and edges, thus showing a strong enhancement. EES mapping indicates a strong EM field enhancement near the tips corresponding to a dipolar mode, whereas for increasing energies other modes start to appear presenting the highest enhancement at the edges and at the center of the NT [49]. It is possible to select the wavelengths of plasmon resonance by changing the aspect ratio (edge/thickness) of the AuNTs so that, as in the case of AuNRs, the LSPR of AuNTs are characterized by a fine tunability. Furthermore, depending on the synthetic procedure, AuNTs display a different grade of truncation that enable the dipolar LSPR tuning, leading to a blue-shift in the spectrum when increasing the snip size of the missing corner [50]. As in the case of AuNRs, the ability to generate great EM field enhancement and the plasmon resonance tunability by means of morphology renders AuNTs an intriguing substrate for SERS analysis. As demonstrated by Tan [51], there is a strong relationship between the SERS enhancement and the LSPR of the corresponding substrate: when laser excitation wavelengths match the LSPR, the SERS signal is found about two orders stronger compared to the case in which the laser excitation wavelength is far away from the LSPR band.

### 4.3. Au Nanostars

Gold nanostars are particles with a star-like morphology comprising a spherical central core from which radial, acute tips branch out, and over the last few years have raised particular attention thanks to their features such as the ease of synthesis for large scale production, the high surface-to-volume ratio useful for improving drug loading efficiency, and above all, thanks to their unusual optical and plasmonic properties [52,53] which pave the way to a great potential for nanomedicine applications. Despite the irregularity of the particle star-like morphology, the extinction spectra show well-defined localized surface plasmon features. The UV-Visible spectra of Au nanostars display an intense band centered at ca. 650–900 nm and a weaker band localized at ca. 500–600 nm [54]. Assigning the LSPR of Au nanostar bands by means of theoretical models for the resolution of the Maxwell’s equations, such as Finite-Difference Time-Domain (FDTD) [52], Discrete Dipole Approximation (DDA) [55] or Boundary Element Method (BEM) [54], produced the same results: the band at lower energy is assigned to dipolar resonances localized at the individual tip, whereas the band localized at higher energy is assigned to dipolar resonances localized at the central core. Interestingly, FDTD calculations have demonstrated, despite the general observation that the UV-Visible spectra are dominated by the LSPR band associated with tip oscillations, that the plasmon modes of Au nanostars arise from the hybridization of resonances associated with the core and the tips generating bonding and antibonding nanostar plasmon. Thus, a contribution from the core plasmon to the tip plasmon is proposed. The core plasmons have larger frequencies than the tip plasmons, so that the conduction electron of the core structure can adiabatically follow the lower frequency tip plasmon oscillations. This results in an “antenna effect” which is responsible for an increase in the extinction cross-section (a factor of 4-fold with respect to the individual tip plasmons) as well as in the electric field enhancement. In Au nanostars, the central sphere acts as an electron reservoir. Regarding the influence of the specific morphological details, it has been found that the aperture angle and the roundness of the tip are of major importance in affecting the energy of the LSPR tip mode and thus its position in the UV-Visible spectra, since small changes lead to a significant shift in the main band. BEM calculations have been confirmed when investigating the spatial distribution of the plasmon modes of gold nanostars by electron energy-loss spectroscopy (EELS) mapping performed on a single particle in a scanning transmission electron microscope (STEM) [56], which also showed a high localization near the tips. Similar to Au nanorods, the plasmon resonance in AuNSs can be tuned through modifications in the aspect ratio and/or in the tip sharpness. When the aspect ratio increases, it is possible to observe a red-shift in LSPR, and larger Au nanostars present an increased number and longer tips along with a red-shift in plasmon resonances at an increase in AuNS size. Despite the great potential, understanding how the morphology can impact the efficacy of AuNSs as a field enhancer is a controversial task. In an early work, Vo-Dinh [57] investigated the properties of AuNSs as SERS substrates and compared with respect to the size. Changing the ratio between seeds and HAuCl_4_, PVP-functionalized AuNs with sizes in the range from 45 nm to 116 nm have been synthesized. Even though the morphology of AuNSs with distinct sizes is different, exploiting p-mercaptobenzoic acid (p-MBA) as a probe molecule, EF around 5 × 10^3^ has been achieved, thus revealing no great differences among the different sizes. Nevertheless, from the results obtained by Ganesh [58] it has been found that when the size and length of the spikes of Au NS have been changed, the intensity of the SERS signal is significantly altered. Exploiting AuNSs with two different morphologies, short spike (SSNS) and long spike (LSNS) Au nanostars, from the SERS experiment demonstrated that SSNS exhibit a higher SERS enhancement compared to LSNS, and it has been justified with a size effect. According to the work of Hong and Li [59], the optimized size of Au nanoparticles for obtaining maximum enhancement in Raman signal is around 50 nm and the core size of SSNS falls within this regime exhibiting higher intense Raman signals. Regardless of the controversy in correlating the AuNSs morphology to SERS activity, their own optical properties such as the strong near field enhancement and plasmon resonance tunability enable Au nanostars to be employed in SERS analysis with great potential.

In Figure 2, the electromagnetic field around a gold nanoparticle with a different morphology shown.

With the aim to evaluate how the nanoparticles’ morphology can affect the SERS activity, Kundu, in his work [60], used rhodamine 6 G (R6G) as probe molecule exploiting nanoparticles with different shapes (Au nanospheres, Au nanorods, Au nanowires and Au nanoprisms) as SERS substrates. Calculating the enhancement factor (EF) according to Equation (1), it has been observed that Au nanoprisms showed much higher value of EF compared to other shapes as follows: nanoprisms > nanowires > nanorods > nanospheres. The enhanced SERS activity for nanoprisms has been related with the presence of a greater number of edges: the largest electric field is localized near the sharpest surface or at the sharp ends of the NPs so that Raman enhancement reaches its maximum value at the sharpest surface and, in addition, edges can interact strongly with R6G compared to the smooth surface of the nanospheres. The synergistic effect of EM field enhancement and strong interactions with probe molecules results in the highest EF values. Similar results have been obtained by Wu [61]. The effect of particle shape has been investigated choosing three types of Au nanoparticles (nanospheres, nanorods and nanostars) as SERS substrates and using malachite green isothiocyanate (MGITC) molecules as probes. From the spectra collected under excitation from the 532 nm and 785 nm lasers, it was shown that Au nanostars displayed the highest SERS enhancement factor (EF) while the nanospheres possessed the lowest SERS activity under excitation with 532 and 785 nm lasers. The experimental results have been combined with theoretical calculations in order to provide a deep insight into the relationship between the particle morphology and its SERS activity. The field distribution has been investigated by means of finite-difference time-domain (FDTD) simulation, revealing considerable differences in the distribution of the EM field around the diverse nanoparticle types induced by their localized surface plasmon resonance (LSPR) under both 532 and 785 nm incident lasers. From FDTD simulations, it can be seen that the maximum electric field intensity is concentrated around the sharp tip in the anisotropic structure generating ‘hot spots’ thus leading to the dominant contribution to the SERS intensity. According to the particle morphology and anisotropy, the Au nanoparticles follow the order nanostars > nanorods > nanospheres in terms of enhancement factors. The size and morphology of plasmonic metal nanostructures have been discussed as key factors in determining the formation of the hotspots’ network in a SERS substrate, but other factors, such as their spatial arrangement and nanogap distances need to be controlled in order to obtain optimal plasmonic properties and maximal Raman signal amplification. When plasmonic metal nanostructures are connected, a direct exchange of free electrons is possible producing different types of plasmons, such as the longitudinal antenna plasmon mode (LAP), bonding dipole plasmon mode (BDP) and charge transfer plasmon mode (CTP) [62]. The type of plasmon created (BDP, CTP or LAP) will affect the performances of the SERS substrate. The particle surface morphology is a further key factor to be considered to control the performances of the substrate sensing. In addition, the shape of plasmonic metal nanostructures has been engineered to obtain an enhanced EM field that is beneficial to the SERS analysis, but it is worth mentioning other approaches aiming to obtain promising results in this direction and that are based on the combination of LSPRs with propagating surface plasmons (PSPs). Often, surface plasmons (SPs) are classified in two categories: PSPs, that are running surface waves, and localized surface plasmons (LSPs), collective surface charge oscillations in the form of the standing waves confined in a metal nanoparticle [63]. The incident light cannot easily excite the PSPs and in order to achieve SPRs’ excitation, a specific configuration is required and needs to be assembled [64]. As opposed, the LSPRs are excited by directly shining the light on an assembly of plasmonic metal nanoparticles. Very recently, it has been found that the enhancement factors in SERS analysis are greatly enhanced when the excitation of LSPs is produced by PSPs generated as surface EM waves, and excited from a special coupling medium (prism, waveguide, fiber, or grating) [65], thus paving the way to the development of new devices.

## 5. Effect of Plasmonic Metal Particle Morphology in Hybrid Compounds on SERS

Taking into account the effect of particle size on the SERS efficacy, it has been easy to combine anisotropic nanoparticles with bidimensional material to further improve the SERS efficacy of the hybrid composites thus exploiting the engineering of particle morphology by tailoring the particle anisotropy along with the plasmonic properties and intrinsic electromagnetic ‘‘hotspots’’. In Figure 3, a schematic of a hybrid compound formed by a graphene layer and an Au nanorods with SERS signal is represented.

In the work of Liu [66], for the first time, the fabrication of a hybrid composed of AuNRs and GO for SERS (GO–AuNR) analysis has been reported. The GO–AuNR composite material has been developed exploiting an electrostatic self-assembly strategy and then tested as a SERS substrate. To test the SERS efficacy of the composite system, the Raman signals of four dye molecules (crystal violet (CV), neutral red (NR), trypan blue (TB) and ponceau S (PS)) on different substrates (SiO_2_/Si, AuNRs, GO and GO–AuNR) have been compared. From the results obtained, it has been possible to observe that strong SERS activity is observed when CV and NR are deposited on the GO–AuNRs substrate whereas no SERS effect has been detected for TB and PS dye molecules. In the case of cationic dye (CV and NR), the molecules are electrostatically attracted by the negatively charged GO–AuNRs substrate, whereas the anionic dye TB and PS due to their negative charge are prevented to interact with the substrate, thus weaking the Raman signal. Although exploiting hybrid composites containing AuNRs has revealed improved performances in SERS analysis, the substrate is still not properly adequate for detecting real samples, that can be negatively charged. Further improvement has been reported in successive works. For example, Jiang et al. [67] fabricated a composite containing aggregated Ag nanorods and GO showing good stability and used as a SERS platform to detect Rh6G molecules and trace I^−^ ions in the solution exploiting the SERS quenching due to the formation of the Rh6G-I complex thus finding a LOD of 0.2 nmol/L for Rh6G and a LOD of 0.004 μmol/L. A different strategy exploiting core–shell Au@Ag nanorods has been developed by Gao et al. [68] to obtain an advanced SERS substrate. Using Au@Ag core–shell nanorods hybridized with reduced graphene oxide (GO-Au@AgNRs) Rh6G molecules have been detected with an enhancement factor (EF) up to (5.0 ± 0.2) × 10^8^, 4-fold higher compared to that obtained with rGO-AuNRs and pesticide thiram with a limit of detection (LOD) of 5.12 × 10^−3^ μM has been revealed. An early hybrid system containing Au nanotriangles has been developed by Jiang [69]. Reduced graphene oxide/silver nanotriangle (rGO/AgNT) composite sol was prepared by the reduction of silver ions with sodium borohydride in the presence of H_2_O_2_ and sodium citrate and exploited for SERS detection of Dopamine (DA). The detection is based on the competitive adsorption occurred between DA and the molecular probe acridine red (AR) onto the reduced graphene oxide (GO) nanosheets. Depending on the formation of multiple hydrogen bonding and π-π stacking, DA molecules display a much stronger affinity towards the GO nanosheets compared to AR molecules, so that, when added to the system, DA molecules competed with AR for similar adsorption sites on the rGO surface thus leading to the desorption of AR molecules from the rGO surface. The desorbed AR molecules can be successively adsorbed on the surface of Au nanotriangles thus enhancing the SERS signal: when increasing the DA concentration, the amount of AR molecules adsorbed onto AuNTs rises with SERS intensity responding linearly with DA concentration. Stimulated by intense local electric field enhancements of silver nanoplatelets (Ag-NPs) caused by the anisotropic morphology and sharp corners compared with other morphologies, such as nanospheres, Meng [70] developed hybrid systems containing AgNPs and graphene nanosheets (Ag-NP@GH) to be exploited as SERS substrates taking advantage of their unique properties. The SERS enhancement capability of the Ag-NP@GH composite has been tested calculating the enhancement factor (EF) using rhodamine 6G (R6G) as a probe molecule and a value of 4.7 × 10^8^ for EF has been found, thus confirming the good performance of Ag-NP@GH in SERS analysis. After testing, the composite has been exploited for organic pesticide detection, including thiram and methyl parathion (MP), and their mixtures finding a LOD of 40 nM for thiram and 600 nM for MP. The improved sensitivity of the Ag-NPs@GH results from the combination of the EM effect of the AgNPs and the CM effect of the graphene. Hotspots are generated by AgNPs staying close to each other, whereas graphene nanosheets contribute to SERS efficacy with a chemical enhancement (CM) due to the strong adsorption capability by means of π-π interactions. Aiming to exploit the unique plasmonic properties of gold nanostars (AuNSs), in the work of Krishnan [71] hybrid systems containing graphene oxide (GO) and AuNSs have been produced and used as SERS-active substrates. A simple and eco-friendly synthetic route based on a deep eutectic solvent (DES) has been developed and evaluated as a SERS substrate using crystal violet (CV) as a probe molecule and a value of 1.7 × 10^5^ for the enhancement factor EF and a limit of detection (LOD) of 10^−11^ M have been found. The improved performances of the composite as a SERS substrate have been explained with the large number of nanogaps between two contiguous AuNSs generating the SERS hotspots and to its morphology, permitting to the CV analyte molecules to diffuse inside the structure. One of the early hybrid structures comprising branched Au nanoparticles was developed by Ray in his work [72] in a four-step process, binding the Au-branched nanoparticles, (termed Au nanopopcorns) by means of Cysteamine molecules on graphene oxide (GO) nanosheets. To evaluate the SERS enhancement capability of the composites, SERS spectra were collected using Rh6G dyes as probe molecules and GO, Au nanopocorns and hybrid nanostructures as substrates. Calculating the enhancement factor (EF), it has been found that the higher value of EF is for the composites, followed by Au nanopopcorns and GO. The signal enhancement in the case of GO has been explained with the chemical mechanism, whereas the effect of nanoparticles has been further investigated. When exploiting hybrid composites containing differently shaped (spherical, cage and popcorns) Au nanoparticles, the highest EF for the popcorn shape has been found, depending on the presence of sharp tips. As well as in combination with GO, the differently shaped Au nanoparticles showed the highest EF for nanopopcorns. The hybrid structures containing Au nanopopcorns revealed the best performances in the HIV-1 gag-gene DNA and Staphylococcus aureus (MRSA). Animated by the intriguing properties of hybrid structures, Li [73] developed a SERS substrate for bilirubin detection integrating composites containing AuNS-decorated graphene oxide (GO) nanosheets on common filter paper. The AuNSs/GO hybrids have been assembled by means of electrostatic interactions created by a deposited layer of Poly (diallyldimethyl ammonium chloride) (PDDA) on GO nanosheets and then exploited for bilirubin detection. The resulting SERS substrate combines the EM effect originated by hotspots generated from AuNSs and the ability of GO nanosheets to adsorb bilirubin molecules by means of strong electrostatic and π-π interactions as shown by kinetics measurements. In addition, the SERS performances are improved by the superquenching of fluorescence by both GO nanosheets and GNSs. The SERS substrate showed an LOD as low as 0.436 μM for free bilirubin in blood serum, thus holding considerable properties for clinical translation in accurate diagnosis of jaundice and its related diseases. In Table 2, the performance of hybrid compounds containing metal nanoparticles with different morphologies is reported.

The effect of nanoparticle morphology has been reviewed in this section, but it is noteworthy to mention that in other works, structures formed by plasmonic metal nanohole arrays hybridized with graphene have been exploited with promising results in SERS analysis [74,75].

## 6. Engineered 2D Nanomaterials

Tailoring the anisotropy in nanoparticles morphology amplifies the SERS signal by means of EM field intensification near the nanoparticles; nevertheless, the SERS enhancement also relies on the number of the hotspots created by the structure used as a substrate, so that alternative approaches optimize the hybrid composite structure in such a way as to create the greatest possible number of hotspots by engineering the bidimensional material according to different strategies. Some strategies, for example, exploited the intrinsic properties of the bidimensional material to amplify the SERS efficacy. Recently, in the work of Su [76], a SERS substrate composed of 1T-MoS_2_ nanosheets decorated with silver nanocubes (1T-MoS_2_/AgNCs) and assembled on filter paper has been developed and used for thiram (TRM) and thiabendazole (TBZ) residues in apple fruit detection. The two different phases of MoS_2_, the 1T (metallic, trigonal) and 2H (semiconducting, hexagonal), have been used to create the hybrids with Ag nanocubes (1T-MoS_2_ /AgNCs and 2H-MoS_2_/AgNCs) and tested in SERS analysis with the model molecules rhodamine 6G (R6G) in order to evaluate the effect of the MoS_2_ phase on the SERS performances. The best performances obtained for the composites containing the metallic 1T phase have been explained with a superior electron transfer from Ag to 1T-MoS_2_ compared to the 2H-MoS_2_ phase, depending on the absence of band gap, the lower binding energies of 1T-MoS_2_ compared to 2H-MoS_2_ and the more abundant density of state (DOS) near the Fermi level. In the work of Koratkar [77], a SERS substrate composed of monolayer MoS_2_ decorated with AuNPs has been engineered to obtain higher SERS analysis performances. By means of low-power focused laser cutting, artificial edges have been sculpted in monolayer MoS_2_, on which AuNPs, when deposited by drop-casting, tend to predominantly accumulate. The huge density of AuNPs along these artificial edges concentrates the plasmonic effects in this region, so that hotspots are generated exclusively along these artificial edges. Calculations of first-principles density functional theory (DFT) suggested that AuNPs are strongly coupled to the artificial edges through dangling bonds that are widespread along the unpassivated edges cut by the laser. Moreover, according to DFT calculations, as a result of AuNP binding, there is an enriched availability of conduction channels around the Fermi level so that artificial edges decorated with AuNPs displayed a higher electrical conductivity. The dense assemblage of AuNPs and the increased electrical conductivity generate along the artificial edges regions of mobile charge oscillating in phase with the laser light that drastically enhanced the magnetic field and the SERS response. Using Raman mapping it has been possible to localize the hotspots along the MoS_2_ edges cut by the laser. Inspired by the intrinsic properties of boron nitride (BN) nanosheets, Li et al. [78] have developed a SERS substrate composed of faceted Au nanoparticles synthesized over BN nanosheets by a simple sputtering and annealing method. The stronger resistance to oxidation renders more advantageous the use of BN nanosheets as reusable SERS substrates as they support the heating at high temperatures in air necessary to remove the analyte molecules for reusing. Furthermore, different from graphene which introduces intrinsic Raman band of high intensity in SERS spectra [79], BN nanosheets only display a Raman G band [80] of a low intensity that are barely enhanced by AuNPs, so that interferences are not created and only Raman signals from analytes are shown. The performances in SERS analysis have been tested using rhodamine 6G (R6G) as the probe molecule and silicon oxide (SiO2/Si), atomically thin BN and bulk hBN substrates decorated with AuNPs obtaining the greater enhancement signal for structures comprising BN nanosheets. Even though AuNPs are able to adsorb a certain quantity of R6G molecules, as a result of π-π interactions the BN surface can adsorb a drastically higher number of molecules that are localized in the hotspots between AuNPs, thus enhancing the SERS signal. As expected on the basis of the stronger resistance to oxidation of BN, the experimental results confirmed the ability of hybrid composites to be reused as SERS substrates, being able to sustain multiple thermal regeneration cycles. The performances in SERS analysis for hybrid compounds containing engineered 2D nanomaterials are reported in Table 3.

## 7. Three-Dimensional Structures

Despite the huge steps forward, the development of the SERS platforms is still mainly limited to plasmonic metal nanoparticles hybridized with bidimensional nanomaterial, but three-dimensional (3D) nanostructures start to be used to create hybrid materials. In his work [81], Yin prepared three-dimensional MoS_2_ nanohybrids according to the microwave irradiation hydrothermal synthesis strategy (3D MoS_2_-NS@Au-NPs) and the system created has been compared with bidimensional MoS_2_/Au nanoparticle hybrids and tested as a SERS platform for melamine in milk detection. From comparison, it can be seen that the SERS activity of 3D MoS_2_-NS@Au-NPs structures is improved by almost 56.4-fold in EF compared to 2D MoS_2_-NST@Au-NPs hybrid structures. The amplification of EF has been ascribed to the larger surface area for adsorbing probe molecules and the higher number of hot spots generated to benefit the SERS performances supplied by the three-dimensional structure. Once optimized, the effectiveness of the generated hot spots by tuning the size and the density of the Au nanoparticles of the composite, the optimized structures have been tested for melamine quantitative detection in milk, finding a LOD of 1 ppb, a value lower than the maximum level of melamine as 2.5 ppm in food prescribed by the U.S. Food and Drug Administration. In the same year, Wang et al. [82] fabricated hierarchical MoS_2_-microspheres (MoS_2_-MS) decorated with “cauliflower-like” AuNP arrays (CF-AuNPs), by means of a new synthetic route, which have been successively investigated and tested for SERS analysis. According to the obtained results, it is possible to tailor the average size of CF-AuNPs@MoS_2_-MS nanocomposites by tailoring the molar ratio between MoS_2_-MS and HAuCl_4_ and, once optimized to achieve the best performances in SERS analysis, they have been tested for molecular detection. In addition to R6G and methylene blue (MB) molecule sensing, that showed an LOD, respectively, of 10^−14^ M and 10^15^ M, the composites have been tested for the detection of various metabolites in human early morning urine with promising results. Furthermore, the CF-AuNPs@MoS_2_-MS nanocomposites have been inserted in cellulose acetate membrane (CAM) to fabricate flexible wafer-scale flexible SERS substrates. From the results obtained by exploiting three-dimensional nanostructures hybridized with metal plasmonic nanoparticles as substrates in SERS analysis, it has been shown that the main advantage of these composites resides in their huge surface area. In fact, due to their large surface area the composites can adsorb a higher number of probe molecules compared to composites containing bidimensional materials and, in addition, are able to generate hotspots distributed in the space that are more efficient in enhancing the SERS signal by means of electrochemical mechanism. The effect of surface area in SERS analysis efficacy has been explored by Singh [83]. By means of facile hydrothermal method, a series of MoS_2_ nanoflowers with a surface area ranging from 5 m^2^/g to 20 m^2^/g has been synthesized and tested in SERS analysis using the R6G as a probe molecule. It has been found a linear dependency of SERS signals originating from different substrates with the surface area, thus correlating the surface area of the composites with the intensity of the Raman signal. In Table 4, the performances of the hybrid compounds with a three-dimensional structure are reported.

## 8. Nanospacers

Other approaches introduced bidimensional nanomaterial as a nanospacer between layers of plasmonic structures in order to create dense three-dimensional hotspots that support the striking SERS enhancement. In the early work of Zhu [84], the interactions between light and the sub-nanometer gap were systematically investigated. With a simple fabrication technique, a structure composed of graphene sandwiched between two layers of vertically stacked Au NPs has been developed and investigated as a SERS substrate. Performing numerical simulations based on the Finite element method (FEM) to investigate the effect of graphene in the sandwich structure, it has been found that the electric field is strongly amplified in the gap defined by the graphene film between two vertically stacked layers of AuNPs, leading to electric field enhancement of up to 88 times, much higher compared to that of 14 times in the horizontal gaps between Au nanoparticles without graphene. In addition, by changing the number of graphene films it is possible to control the nanogap between the vertical AuNPs layers, and, as consequence, the SERS enhancement. To investigate the effect of the gap induced by a different number of graphene layers, composites of two vertical layers of AuNPs containing a various number of graphene film have been created and used as a substrate in SERS analysis, obtaining the best performance for structures containing graphene monolayer, thus deducing that the coupling between the layers of plasmonic structures decays exponentially with their distance. Using the composite containing the graphene monolayer (4 nm Au/1LG/4 nm Au) as a SERS substrate, an LOD of 10^−9^ M for rhodamine B (RhB) has been found, hence showing a higher sensitivity compared to 4 nm Au/4 nm Au films, that showed an LOD of 10^−7^ for RhB molecules when used as substrates. The same results have been obtained for R6G molecules, with composites containing graphene resulting in a higher sensitivity. The use of 4 nm Au/1LG/4 nm Au in practical applications has been tested on by means of Sudan III and methylene blue detection, finding an LOD of 0,1 nM, thus showing a potential use in areas of food safety, medical diagnostics, biological imaging and environmental pollutant detection. Similar results showing the extraordinary performances in SERS enhancement obtained by exploiting graphene as a nanospacer have been obtained in the work of Man [85]. In this work, Au nanoparticles, (AuNPs), silver nanoparticles (AgNPs) and graphene have been combined in order to form a sandwiched structure, AgNPs/graphene@AuNPs, to be exploited in order to achieve unique performances in SERS analysis. By means of rhodamine R6G and crystal violet (CV) molecules, the composite has been experimentally tested in SERS analysis generating a huge amplification in Raman signal intensity. The excellent signal SERS enhancement obtained has been explained as the combination of chemical mechanism (CM) and electromagnetic mechanism (EM). The graphene film induced the CM, thus enhancing the SERS activity, but can also act as a nanospacer able to control the hot spots’ size by changing the number of graphene layers. In fact, the electromagnetic enhancement was the result of three-dimensional hotspots generated by lateral nanogaps (AuNPs-AuNPs, AgNPs-AgNPs) and vertical nanogaps (AgNP-AuNPs), tunable by graphene layers. To investigate the effect of the graphene film, different composites containing a different number of graphene layers have been explored using R6G molecules as probes, finding that the graphene bilayer offers the best performances. A single layer of graphene induced the plasmon tunneling phenomenon due to the short distance (<0.5 nm) between Ag and Au nanoparticles which reduces the plasmonic coupling effect weakening the Raman signal, whereas for higher numbers of graphene layers, the nanogap also increases so that the electromagnetic enhancement is reduced because the enhanced local electric field will exponentially decay with distance. This composite system AgNPs/graphene@AuNPs has been tested for Malachite green (MG) detection in sea water finding an LOD of 10^−11^ M, thus demonstrating a potential ability in practical applications. Motivated by the few studies exploiting WS_2_ bidimensional nanomaterial in SERS analysis and on the presumption that, due to its structure, WS_2_ could promote both chemical enhancement (CM) by means of charge transfer between substrate and probe molecules, and electromagnetic enhancement (EM) by means of the strong coupling between WS_2_ and metallic nanostructures through surface plasmon excitation, thus enhancing the SERS signal, Jiang in his work [86] exploited bidimensional WS_2_ nanomaterial as a nanospacer in hybrid nanostructures. A remarkable SERS platform based on AuNPs/WS_2_@AuNPs nanohybrids has been designed and developed in a multi-step process. Firstly, annealing an Au film deposited onto a SiO_2_ substrate, a layer of Au NPs has been created. Successively, by means of a thermal decomposition process, a bilayer WS_2_ film has been grown onto the AuNPs surface, and finally, a second layer of AuNPs was deposited onto the WS_2_ film by means of a further annealing, thus obtaining the AuNPs/WS_2_@AuNPs composites. Introducing the bilayer WS_2_ film as a nanospacer between the two layers of plasmonic structures, a highly enhanced local electromagnetic field has been generated. Dense 3D hotspots occurring through this hybrid plasmonic nanostructures are responsible for the greatly enhanced SERS performances. Using rhodamine R6G as a probe molecule to test the performance in SERS analysis, the AuNPs/WS_2_@AuNPs nanohybrids showed an excellent sensitivity with the minimum detectable concentration of 10^−11^ M. In addition, the AuNPs/WS_2_@AuNPs nanohybrids showed extremely satisfying performances in detecting other probe molecules such as crystal violet (CV) molecules. The results obtained in the presented works illustrate the role of bidimensional nanomaterial used as a nanospacer between the layer of plasmonic metal nanostructures in the enhancement of the SERS signal. When used as a nanospacer, the bidimensional nanomaterial is able to create three-dimensional hotspots generated by the combination of lateral nanogaps (gaps inside plasmonic layer) and vertical nanogaps (gaps between plasmonic nanolayers). The great benefit of using a nanospacer is the possibility to finely tune the vertical gap by changing the number of bidimensional nanomaterial layers and taking into account that an exponential decay controls the coupling between two plasmonic layers, so that increasing the nanogap with a higher number of 2D nanomaterial layers reduces the SERS enhancement, but also for too small nanogaps, the Raman signal is weakened by the plasmon tunneling phenomenon. The performances of hybrid structures containing nanospacers in SERS detection are listed in Table 5.

## 9. Bidimensional Nanomaterials Used to Veil AuNP Arrays

In the majority of the hybrid composites that have been developed, the bidimensional nanomaterial acts as a support for the plasmonic metal structures; however, this configuration only provides a limited number of contact points between 2D nanomaterials and metal nanostructures: to attain a large enhancement of the SERS signal, an efficient contact between the metal framework and the bidimensional nanomaterial is necessary [87]. On this basis, different strategies wrapped the 2D nanomaterial around plasmonic metal nanoparticles in such a way that the number of contact points is increased, thus optimizing the Raman enhancement. A SERS substrate has been created veiling an array of silver nanocubes (AgNCs) with a graphene oxide (GO) film by means of a simple GO deposition process based on GO self-assembly on the metal surface [88]. According to the finite element method (FEM) calculations, the maximum intensity is 70% reduced when a 7 nm-thick GO layer is added on the metal structure surface and a more spread E-field distribution along the cluster edges and at the interface between particles is generated after supporting a 7 nm-thick GO layer on AgNCs, in contrast with well-localized hot spots observed for bare cubes. In conformity with FEM calculations, a reduction in SERS efficacy is expected, but the experimental results pointed out a superior SERS activity including more resolved peaks with higher signal intensity and larger reproducibility. The excellent SERS activity has been explained with a chemical enhancement deriving from a combination of π-π interactions and charge transfer from the oxygen-rich functional groups of GO to the probe molecules and in addition, with the GO ability to catch different compounds that therefore are accumulated on its surface thus intensifying the SERS signal. Comparing the performances, it has been found that wrapping plasmonic nanostructures enables a greater enhancement of the SERS signal than supporting on the bidimensional material. Intrigued by the great potential of Au nanoparticles hybridized with graphene nanosheets as SERS substrates, Cerruti [89] produced Au nanostars wrapped by graphene oxide (GO) nanosheets which further improved the SERS platform. Previously synthesized AuNSs have been functionalized with the positively charged Cysteamine, that create electrostatic interactions with negatively charged GO, so that GO-wrapped AuNSs (Au NSt@nGO) have been produced and tested in SERS analysis. As a result of AuNSs being wrapped with GO, the Raman signal of nGO by 5.3-fold compared to samples in which nGO is in contact with the nanostars but does not wrap them, whereas there is a higher enhancement for wrapped AuNSs, thus confirming the efficiency of wrapping to improve the SERS signal. SERS signals of typical Raman reporter such as rhodamine B (RhB), crystal violet (CV) and R6G sandwiched between AuNSs and GO nanosheets were higher compared to the signal obtained when the molecules are adsorbed on the nanostar surface. Together with an increase in SERS efficiency, wrapping AuNSs with GO results in a greater physiological stability, depending on a prevented RhB desorption in physiological conditions. A more detailed interpretation of SERS efficacy generated by plasmonic nanoparticle wrapping with a bidimensional material is given in the work of Chen [90]. Inspired by the strong surface adsorption of airborne hydrocarbon and aromatic molecules of thin boron nitride (BN) nanosheets, a SERS platform has been created placing an atomically thin BN nanosheet over an Au nanoparticle array produced via physical processes. Using R6G as probe molecules and BN nanosheets with different thicknesses, it has been found that Raman signals were most prominent for the lower thickness of BN, but reduced when the layer thickness increased and the stronger Raman signals were attributed to hotspots. Thinner BN nanosheets, due to their greater flexibility are able to better conform to the underlying AuNPs so that the analyte molecules were closer to the plasmonic hotspots. For increasing thickness, the BN nanosheets were much less deformed so that analyte molecules were more distant from the plasmon-induced EM field which decays exponentially with the distance. In addition, more R6G molecules were attracted depending on the strong adsorption ability of BN nanosheets towards aromatic molecules by means of π-π interactions. Conformational changes explain the stronger adsorption capability, with BN nanosheet polarity not contributing to such effect. Wrapping plasmonic metal nanostructures with graphene nanosheets has revealed great potential, thus expanding applications of hybrid composites from SERS detection to SERS bioimaging. Graphene oxide (GO)-wrapped Au nanorods (GO@GNRs) have been created by Wu [91]. Assessing the cytotoxicity showed a greatly enhanced biocompatibility of GO@GNRs, provided by the encapsulation enabling a reduced contact with the surrounding environment, thus decreasing the amount of residual CTAB that induces cytotoxicity. The SERS activity in the near infrared (NIR) has been investigated using six dye molecules as probes showing extremely intense SERS signals and highly enhanced activities of NIR SERS multiple effects, such as Au nanorods LSPR, the charge transfer between graphene nanosheets and probe molecules and the enrichment of dye molecules on the GO sheets. The enhanced NIR SERS activity and the improved biocompatibility enable a successful application of GO@GNRs as a robust nanoplatform for ultrafast NIR SERS bioimaging. On these bases, exploiting bidimensional materials to veil arrays of plasmonic nanostructures produces a greater SERS activity compared to that obtained when 2D material is used as a support. Due to its flexibility, the bidimensional nanomaterial is able to bring analyte molecules into the proximity of the hotspots originated from plasmonic structures thus enhancing the SERS efficacy. Moreover, using 2D nanomaterials as veiling medium offers the additional advantage of protecting the arrays of plasmonic nanostructures, thus increasing the stability of the SERS substrate and broadening the composite range of applications. Wrapping plasmonic nanostructures imparts a biocompatibility to the system, thus that can also be exploited in the biomedical field.

## 10. Future Perspectives

In order to give a better understanding of its enormous potentiality, the future perspectives of the SERS technique will be provided considering a range of directions in which the research in this field is focused. Since its development, SERS has affirmed a great potential as a powerful technique to detect simple and more complex molecules, in contact or adjacent to a plasmonic substrate that generally consists of a metal surface, and recently, also of hybrid materials. Such enormous potential, combined with advances in the development of associated instrumentation has produced an outbreak of research, moving forward many different applications ranging from materials and environmental science through biology and medicine. Some clinical implementations of SERS that could find an application in the near future are: 1) the detection of tumor margins during surgery [92], as it has been demonstrated that tags were sufficient to improve tracking of tumor margins employing a portable Raman microscope instead of a benchtop instrument; 2) in optical fiber-guided imaging procedures such as endoscopy, colonoscopy, or others used to detect and to visualize superficial diseased tissues within the body [93]; and 3) in liquid biopsy, a term widely comprising the identification of disease biomarkers in blood or other bodily fluids [94]. Liquid biopsy holds great potential to simplify disease detection and monitoring and make it less painful for the patient. In particular, the monitoring of the disease progresses and the response to therapy would ideally be allowed on a daily basis, thus avoiding the dependence on imaging approaches that could not detect changes in the size of the tumor. The SERS analysis has also been revealed as a valid tool to detect inorganic and highly toxic organic pollutants as well as to monitor bacterial contaminant with a detection threshold in the parts for a billion range. In food analytics, detection limits in quality control and nutrient quantification down to the nanomolar range have been reached. The difficulties in analyzing the surface residues, an issue of significance in a variety of areas including health and safety, homeland security, forensics, etc., paved the way to the development of flexible SERS substrates, still a young research area within the progress of the SERS technique. The greater part of recent flexible substrates designed for point of care analysis can be classified according to two categories: sticky “SERS tapes” and adsorptive “SERS swabs”. Typically, SERS tapes are adhesive and flexible plastic films, bearing plasmonic particles on the surface, that can be pressed and peeled from the sample surface to extract the molecules for in situ analysis [95], whereas SERS swabs can be exploited to collect chemical compounds by dabbing the surface of the sample [96]. A main obstacle in the present applications of SERS is the difficulty in analyzing complex real samples, containing a large variety of chemical species and micro/macro-contaminants in addition to the target analyte molecules. Often, these impurities are present in much higher concentrations than the analyte, thus they can interfere with the analysis resulting in significantly reduced sensitivity and reproducibility. The method developed to address this issue still requires sophisticated equipment for the pre-treatment of the samples so that the analyses are restricted to a laboratory setting and must be performed by trained professionals. In this scenario, the difficulties in analyzing SERS spectra from complex samples for disease diagnosis and food analytics pave the way to the implementation of Artificial Intelligence [97]: the acquired SERS spectra of such samples could be spectrally unmixed so that the concentration profiles for better quantification could be estimated. The algorithm offers the additional advantages to characterize the disease while evaluating the adulterants and toxins in food processing as well. Thus, while SERS has experienced enormous progress, a broader range of use is limited by high running costs, as well as their inefficacy in point-of-care analysis. In view of the tremendous progress in the implementation of the SERS technique as an analytical tool, a great challenge is a rapid evolution in commercial products such as compact setups, tailored sensing platforms or efficient imaging methods that are able to compete or complete goods in current use in a wide range of technologies. In the next steps, SERS must be developed such that its processes of application will be simplified and its use of routine will be enabled by non-specialists. In light of the commercialization, adequacy for automated mass production and stability during storage must be examined.

## 11. Conclusions

Between the plethora of approaches focused on improving the performances and making SERS analysis a routine technique to enhance the quality of life of people, in recent years the development of sensing platforms based on structures composed of plasmonic metal nanoparticles hybridized with bidimensional nanoparticles have attracted great interest due to their outstanding properties. In this review, the different approaches used to enhance the outcomes of hybrid compounds have been discussed. Initially, the simplest structures based on spherical metal nanoparticles hybridized with graphene and its derivates (the most common bidimensional nanomaterial) have been considered, and then it has been considered how the structures have been evolved by changing the particle shape or engineering the bidimensional material to obtain the highest values in the enhancement factor (EF). Due to a greater number of analyte molecules adsorbed and the number of hotspots formed, three-dimensional (3D) structures of 2D nanomaterials and plasmonic nanoparticles showed the best results, so that, most likely, the future steps will be focused on the development of such compounds. The future perspectives of the technique have been discussed from a broader point of view considering the various directions in which the technique is progressing and not only from a point of view of the hybrid compounds. In this scenario, multifunctional substrates combining several of the characteristics considered will be developed in order to render SERS a daily technique for better living conditions.

## Figures and Tables

**Figure 1 biosensors-12-00225-f001:**
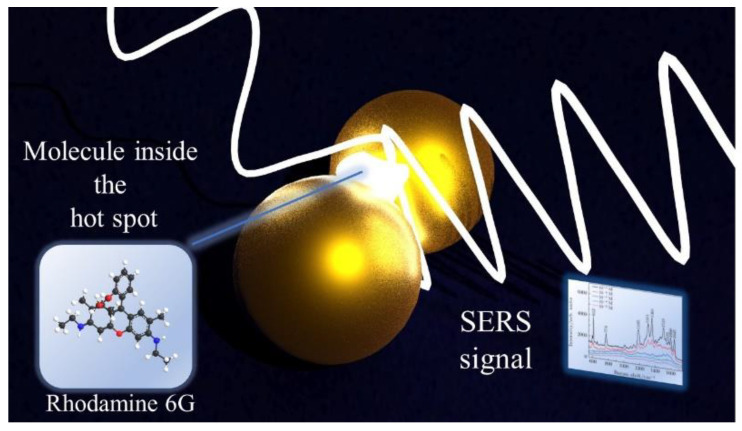
Schematics showing the SERS signal arising from a molecule localized inside a hotspot created in the space between two plasmonic metal nanoparticles.

**Figure 2 biosensors-12-00225-f002:**
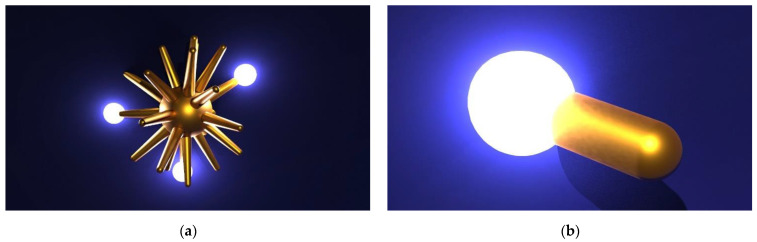
(**a**) Electromagnetic (EM) field (bright spot) around a gold nanostars’ (AuNS) spike and (**b**) EM field around a gold nanorod (AuNR) tip.

**Figure 3 biosensors-12-00225-f003:**
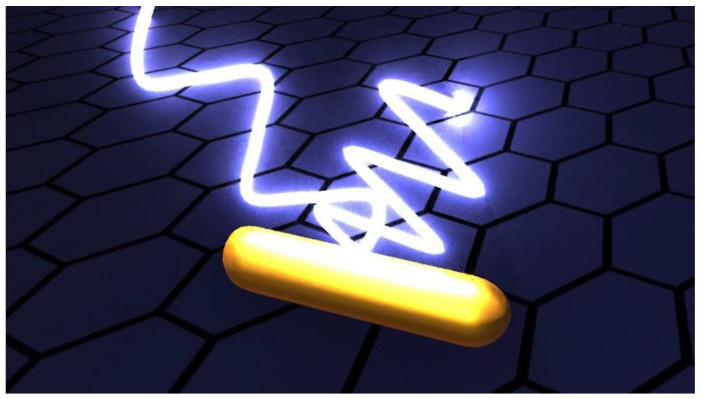
Au nanorods under the illumination of a laser beam and the resulting SERS signal.

**Table 1 biosensors-12-00225-t001:** Performances of the SERS platform based on hybrids of plasmonic metal nanoparticles and graphene or other 2D nanomaterials.

**NANOPARTICLES DEPOSITED ON GRAPHENE**						
**System**	**Molecule Used to Calculate LOD**	**Limit of Detection (LOD)**	**Enhancement Factor (EF)**	**Molecule Used to Calculate EF**	**Equation Used to Calculate EF**	**Reference**
Graphene oxide/Ag nanoparticle hybrids (GO/AgNPs)	Acid folic	9 nM	Not calculated			[35]
Graphene nanosheets/Ag nanoparticle hybrids (Ag/GNs)	2,4,6-trinitrotoluene (TNT)	5 × 10^−16^ M	Not calculated			[36]
Reduced graphene oxide/Au nanoparticle hybrids (RGO/AuNPs)	2-thiouracil (2-TU)	1 μM	5.6 × 10^5^	4-aminothiophenol (4-ATP)	*(I_SERS_/I_bulk_) × (M_bulk_/M_ads_)*	[37]
**NANOPARTICLES GROWN ON DIFFERENT 2D NANOMATERIALS**						
**System**	**Molecule used to calculate LOD**	**Limit of Detection (LOD)**	**Enhancement Factor (EF)**	**Molecule used to calculate EF**	**Equation used to calculate EF**	**Reference**
AuNP-decorated MoS_2_ nanosheets (AuNPs@MoS_2_)	Rhodamine 6G (R6G)	10^−6^ M	8.2 × 10^5^	Rhodamine 6G (R6G)	*(I_SERS_/I_bulk_) × (N_bulk_/N_SERS_)*	[38]
AuNP-decorated tungsten disulfide (WS_2_) nanosheets (Au/WS_2_)	Myoglobin (Mb)	10^−2^ pg mL^−1^	6.78 × 10^6^	Rhodamine 6G (R6G)	*(I_SERS_/I_bulk_) (C_bulk_/C_SERS_) (P_bulk_/P_SERS_)*	[39]

**Table 2 biosensors-12-00225-t002:** Performances of the SERS platform based on different shaped nanoparticles.

**NANORODS**						
**System**	**Molecule Used to Calculate LOD**	**Limit of Detection (LOD)**	**Enhancement Factor (EF)**	**Molecule Used to Calculate EF**	**Equation Used to Calculate EF**	**Reference**
Graphene oxide/Au nanorods hybrids (GO-AuNRs)	Crystal violet (CV), neutral red (NR), blue (TB), ponceau S (PS)	Not calculated	Not calculated			[66]
Silver nanorods/reduced graphene oxide nanosheets hybrids (AgNR/rGO)	Rhodamine 6G (Rh6G)	0.2 nmol/L–0.004 μmol/L	Not calculated			[67]
	Iodine ion					
Au/Ag core–shell nanorods and reduced graphene oxide hybrid structure (Au@AgNRs/rGO)	Thiram	5.12 × 10^−3^ μM	(5.0 ± 0.2) × 10^8^	Rhodamine-6G (R6G))	*(I_SERS_/I_bulk_) × (N_bulk_/N_SERS_)*	[68]
**Au NANOTRIANGLES**						
**System**	**Molecule used to calculate LOD**	**Limit of Detection (LOD)**	**Enhancement Factor (EF)**	**Molecule used to calculate EF**	**Equation used to calculate EF**	**Reference**
Reduced graphene oxide/silver nanotriangles hybrid structures (rGO/AgNT)	dopamine (DA)	1.2 μmol/L	Not calculated			[65]
Ag-nanoplates/graphene hybrids (Ag-NP@GH)	thiram	40 Nm–600 nM	4.7 × 10^8^	Rhodamine 6G (R6G)	*(I_SERS_ × C_0_)/(I_0_ × C_SERS_)*	[66]
	Methyl parathion (MP)					
** A ** **u NANOSTARS**						
**System**	**Molecule Used to Calculate LOD**	**Limit of Detection (LOD)**	**Enhancement Factor (EF)**	**Molecule Used to Calculate EF**	**Equation Used to Calculate EF**	**Reference**
Graphene oxide/Au nanostars hybrid structure (GO/AuNSs)	Crystal violet (CV)	10^−11^ M	1.7 × 10^5^	Crystal violet (CV)	*(I_SERS_/N_SERS_)/(I_Nor_/N_Nor_)*	[69]
Popcorn-shaped gold nanoparticles and graphene oxide hybrid structures	Methicillin-resistant Staphylococcus aureus (MRSA)	10 CFU/mL	3.8 × 10^11^	Rh6G	*(I_SERS_/I_bulk_) × (M_bulk_/M_ads_)*	[70]
Graphene oxide/Au nanostars hybrid structure (GO/AuNSs)	Bilirubin	0.436 μM	2.43 × I_SERS_/I_bulk_	4-nitrothiophenol (4-NTP)	(*I_SERS_/N_SERS_) × (N_bulk_/I_bulk_*)	[71]

**Table 3 biosensors-12-00225-t003:** Performances of the SERS platform based on hybrid compounds containing engineered 2D nanomaterials.

ENGINEERED 2D NANOMATERIAL						
System	Molecule used to calculate LOD	Limit of Detection (LOD)	Enhancement Factor (EF)	Molecule used to calculate EF	Equation used to calculate EF	Reference
Ag nanocube-decorated 1T-MoS_2_ nanosheet composites (1T-MoS_2_/AgNCs)	Thiram (TRM)	0.62 Nm–50 Nm	1.78 × 10^7^	Rhodamine 6G (R6G)	(I_SERS_ × C_Raman_)/(I_Raman_ × C_SERS_)	[76]
	Thiabendazole (TBZ)					
Gold nanoparticle-decorated MoS_2_ nanosheets (n-MoS_2_@AuNP)	R = Rhodamine B (RhB)	10^−10^ M	∼10^4^	R = Rhodamine B (RhB)	Peak intensity ratio of the SERSactive regions and the flake surface	[77]
Gold nanoparticles decorated boron nitride (BN) nanosheets (Au/BN)	Rhodamine 6G (R6G)	5.12 × 10^−3^ μM	(5.0 ± 0.2) × 10^8^	Rhodamine 6G (R6G)	(*I_SERS_/I_bulk_) × (N_bulk_/N_SERS_*)	[78]

**Table 4 biosensors-12-00225-t004:** Performances of the SERS platform based on hybrid compounds with a three-dimensional structure.

THREE DIMENSIONAL STRUCTURES						
System	Molecule Used to Calculate LOD	Limit of Detection (LOD)	Enhancement Factor (EF)	Molecule Used to Calculate EF	Equation Used to Calculate EF	Reference
Gold nanoparticle-decorated three-dimensional (3D) MoS_2_ nanospheres ((3D MoS_2_-NS@Au-NPs)	Melamine	1 ppb	7.9 × 10^7^	4-mercaptophenol (4-MPH)	(*I_SERS_/N_ads_)/(I_bulk_/N_bulk_*)	[81]
Hierarchical MoS_2_-microspheres decorated with “cauliflower-like” AuNP arrays (CF-AuNPs@MoS_2_-MS)	Rhodamine 6G (R6G)	10^−14^–10^−15^	Not calculated			[82]
	Methylene blue (MB)					
Gold nanoparticle-decorated boron nitride (BN) nanosheets (Au/BN)	Rhodamine 6G (R6G)	5.12 × 10^−3^ μM	(5.0 ± 0.2) × 10^8^	Rhodamine 6G (R6G)	(*I_SERS_/I_bulk_) × (N_bulk_/N_SERS_*)	[78]

**Table 5 biosensors-12-00225-t005:** The performances of the SERS platform based on hybrid compounds engineered with nanospacers are listed.

**NANOSPACERS**						
System	Molecule Used to Calculate LOD	Limit of Detection (LOD)	Enhancement Factor (EF)	Molecule Used to Calculate EF	Equation Used to Calculate EF	Reference
Graphene sandwiched between two layers of vertically stacked Au NPs (Au NP/graphene/Au NP)	Sudan III	0.1 nM	1.6 × 10^8^–2.5 × 10^8^	Rhodamine B (RhB)	(*I_SERS_/I_bulk_)/(N_bulk_/ N_SERS_*)	[84]
	Methylene blue			Rhodamine 6G (R6G)		
Graphene nanosheet sandwiched between a layer of AuNPs and AgNPs (AgNPs/graphene@AuNPs)	Malachite green (MG) in deionized (DI) water	10^−11^ M–10^−8^ M	Not calculated			[85]
	Malachite green (MG) in sea water					
WS_2_ nanosheets sandwiched between two Au nanoparticle layers (AuNPs/WS_2_@AuNPs)	Rhodamine 6G (R6G)	10^−11^ M	Not calculated			[86]

## Data Availability

Not applicable.

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
