# Peer review of "Plasmonic Metal Nanoparticles Hybridized with 2D Nanomaterials for SERS Detection: A Review"

_biosensors, 2022, doi:10.3390/bios12040225_

Round 1

Reviewer 1 Report

  1. In the abstract, it is advised to explain the "unique properties" or "outstanding properties".
  2. If there is a limit for the number of words in abstract, it is advised to add one paragraph at the beginning of introduction to discuss the originality or difference from other reviews in this field. In addition, a preview of the following sections is recommended to keep readers informed, like the effect of shape, morphology, 3D structures, nano-spacers, and etc.
  3. Some English language revision is advised. For example, "because is" in the sentence from line 110 to 112. The sentence from line 155 to 163 is too long and recommended to split into two or three sentence. The sentence from 171 to 172 is advised to revise "In the case of folic acid detection, instead, (to adjust)." The sentence from line 393 to 397 is advised to split into two sentences.
  4. In the first paragraph of "2 hybrid nanocomposites", it is advised to add more details about how it shows intriguing properties. For example, it is advised to add specific number of chemical enhancement or comparison to support your statements.
  5. In sections like 5-8, the number of relevant examples is comparatively small. It is advised to add more to relevant references to support each part.
  6. It is advised to add one section for "conclusion and outlook" to discuss the summary and perspective at the end.
  7. The number of reference in this review is comparatively small. As for references, it is suggested to add more relevant references in the past 5 or 10 years. 

Reviewer 2 Report

The review “Plasmonic Metal Nanoparticles Hybridized with 2D Nano-2 materials for SERS Detection: a Review” by Serafinelli et al. is a very interesting manuscript. The authors have written an in-deep introduction, describing the main physical principles of the mechanisms of SERS enhancement. This makes the manuscript very informative.

My major concern consists in the following. In the very first sentence of the abstract, the authors mention “potential single-molecule sensitivity”, but this is the only place where I could find this term in the manuscript. In fact, the development of novel analytical approaches, which can provide single-molecule sensitivity, is one of the crucial tasks of modern biomedical research, and nanotechnology-based methods (including those utilizing atomic force microscopy (AFM) and electrical molecular detectors) are just the case. In this connection, I recommend the authors to consider the methods, which can provide single-molecule sensitivity, in more detail — either by expanding the introduction, or by introducing a separate section within the manuscript. Please, note, that the list of these methods is not limited to optical ones, but also includes electrical and nanomechanical (AFM-based) methods.

I also would like to give a list of recommendations for the authors to indicate how their manuscript can be improved.

First of all, I recommend the authors to review (at least briefly) more recent publications (published in 2020 and 2021), since I have detected that the content of such publications in the reference list is relatively low.

Secondly, I recommend the authors to check their manuscript for misprints and language ambiguities. I have found several confusing misprints, and some points should be corrected to avoid possible misunderstanding:

  1. 25: expected: Raman spectroscopy.
  2. 30: please, define NP (this is the first mention of this abbreviation). Do the authors mention nanoparticles here?
  3. 132-135: Very long sentence. Please, use commas to emphasize the structure of the sentence, ore re-phrase it to clarify its meaning.
  4. 183-184: expected: … by simultaneous reduction…
  5. 332: expected: … to other shapes as follows: nanoprisms > nanowires > nanorods > nanospheres.
  6. 342: “… it was shown …” instead of “… it is shown…”.
  7. 361: Please, add dot in the end of the sentence.
  8. 399: expected: “with increasing the DA concentration…”

Accordingly, I recommend the authors to revise their manuscript.

Sincerely,

The reviewer

Reviewer 3 Report

The authors have reviewed the physics and working principles of SERS based on different shaped metal nanoparticles. The authors have also summarized on the hybrids compounds as the substrates in SERS platform. Overall, this review can inspire more nanomaterial design ideas for SERS platform. Therefore, I would like to recommend this review to publish in Biosensors. Below are a few suggestions for the authors.

1. In Figure 1, a molecule localized in the hotspot and the figure showed a molecule. Therefore, it’s would be better to indicate the name of the molecule.

2. For a review, a section of “discussion” or “challenges and opportunities” should be provided to reveal the future perspective at the end of the review. Please add a section of discussion.

3. For the sentence “Rod-shaped nanoparticles are nanostructures where one dimension is longer than the other two, so that the term nanorods indicates elongated nanoparticles.”, some references could be cited.

https://doi.org/10.1016/j.jhazmat.2020.124617

4. For the sentence “In this contest, bidimensional (2D) nanomaterials such as graphene and its derivatives, transition metal dichalcogenides (TMDs).....”, some examples of TMDs should be cited to reveal the related materials.

https://doi.org/10.1038/natrevmats.2017.33

Round 2

Reviewer 1 Report

No for the revised version.

Author Response

We thanks the reviewer for the positive comment!

Kind Regards

The authors